# Toward Real-Time GNSS Single-Frequency Precise Point Positioning Using Ionospheric Corrections

Vlad Landa [1] and Yuval Reuveni [2,3,4,5,*]

1   Department of Computer Science, Ariel University, Ariel 4070000, Israel; vlad.landa@msmail.ariel.ac.il
2   Department of Physics, Ariel University, Ariel 4070000, Israel
3   Department of Geophysics, Eastern R&D Center, Ariel 4070000, Israel
4   Astrophysics Geophysics and Space Science Research Center, Ariel University, Ariel 4070000, Israel
5   School of Sustainability, Reichman University, IDC, Herzliya 4610101, Israel
*   Correspondence: yuvalr@ariel.ac.il

**Abstract:** Real−time single−frequency precise point positioning (PPP) is a promising low−cost technique for achieving high−precision navigation with sub−meter or centimeter−level accuracy. However, its effectiveness depends heavily on the availability and quality of the real−time iono-spheric state estimations required for correcting the delay in global navigation satellite system (GNSS) signals. In this study, the dynamic mode decomposition (DMD) model is used with global ionospheric vertical total electron content (vTEC) RMS maps to construct 24 h global ionospheric vTEC RMS map forecasts. These forecasts are assimilated with C1P forecast products, and L1 single−frequency positioning solutions are compared with different ionospheric correction models. The study examines the impact of assimilating predicted RMS data and evaluates the presented approach's practicality in utilizing the IGRG product. The results show that the IGSG RMS prediction−based model improves positioning accuracy up to five hours ahead and achieves comparable results to other models, making it a promising technique for obtaining high−precision navigation.

**Keywords:** DMD; ionospheric TEC RMS forecasts; machine learning

## 1. Introduction

In the growing demand for precise point positioning (PPP) applications, which mainly arises in automotive, agriculture, telecom, and hydrography industries, the single–frequency global navigation satellite system (GNSS) receivers are the low–cost alternatives to dual–frequency localization solutions. The effect of ionized air molecules and the forma-tion of free electrons in the Ionosphere [1–6], which are primarily governed by X–ray [7] and extreme ultraviolet (EUV) solar radiation, plays a crucial role in the transmission and recep-tion of radio waves as well as in navigation systems that pass through this medium [8–11]. Unlike, the dual−frequency receivers, which utilize their second channel in order to resolve the ionospheric path delays, due to the propagation through an ionized medium [6,8,9], the single−frequency GNSS devices require an external source for resolving the ionospheric path delay propagation [12,13], especially for PPP applications.

The study of the Ionosphere, especially the estimation of the vertical total electric content (vTEC), which is the primary parameter monitored for space weather research and measures the extent of ionospheric disruption, has an arising interest in the machine learning scientific community for vTEC predictions [14]. Recently, various machine learning approaches combined with different data types, have been applied in order to study the modeling of ionospheric variability using deep recurrent neural networks [15]; forecasting vTEC with the ensemble of RandomForest, AdaBoost, and XGBoost [16]; and even studying the Alaskan ionospheric irregularities [17].

Currently, operational agencies such as the Center for Orbit Determination in Eu-rope (CODE) [18,19], the European Space Agency/European Space Operations Center

(ESA/ESOC), the Jet Propulsion Laboratory (JPL), and more [20], provide global iono−spheric maps (GIM) products. GIM products have a spatial resolution of 2.5 degrees by 5 degrees in latitude and longitude, respectively [21,22], and contain an ionospheric estimation of the vTEC. In addition, GIM products also constitute a correction source for multi−GNSS real−time single−frequency precise point positioning (RT−SFPPP) [23−25].

The GIM products provided by the aforementioned agencies are released with different time latencies and are typically categorized into five classes: real−time, ultra−rapid, rapid, final, and predicted. For example, the Chinese Academy of Sciences (CAS) and the Centre National d'Etudes Spatiales (CNES), which provide real−time GIM products [26,27], as well as the International GNSS Service (IGS) [28,29], which is the most commonly used, provide ionospheric corrections with latencies ranging from 3 h for the ultra−rapid, 17 h for the rapid, and 13 days for the final products. In general, all categories, except the predicted and the real−time, also include the root−mean−square (RMS) maps as the standard deviations of the corresponding vTEC errors for each GIM product.

The optimization of PPP stochastic model algorithms often relies on the GIMs RMS maps as a standard accuracy indicator [30]. As in PPP, obtaining accurate information regarding the ionospheric state, using realist models, is crucial for selecting the optimal weighting strategy, reducing convergence time, and preventing anomalies in positioning errors; the RMS information can also aid in enhancing the overall positioning accuracy of PPP [11]. Therefore, the ability to provide such statistics is substantial for PPP applications, specifically for real−time localization demand.

In the context of RT−SFPPP, the publicly available rapid and final ionospheric correction models [20] share a common time latency disadvantage, meaning that the use of these sources implies accuracy degradation due to the temporal difference and necessity of the estimation process [30]. The available alternative for time−delayed rapid and final products is the predicted GIMs provided by the CODE agency [18,19]. CODE offers predictions for 24 h (1 day) and 48 h (2 days) for vTEC GIMs with a temporal resolution of 2 h, for both predicted temporal ranges. Although the predicted CODE ionospheric correction models constitute a solution for the time latency disadvantage, these products, unfortunately, lack the RMS statistics and, thus, are less suitable for PPP applications.

Therefore, in this study, we propose a data−driven approach, to augment the available predicted one−day CODE products with RMS statistics. The proposed methodology is based on the dynamic mode decomposition (DMD) dimensionality reduction algorithm, which utilizes the reference RMS maps derived from two main sources, the IGSG final solution RMS (IGSG−RMS) and the IGSG rapid product RMS (IGRG−RMS), as a training data set. During the first step, we establish the baseline accuracy of the method by assessing the population of RMS statistics using the IGSG−RMS source. Then, in order to identify the practical implementation of this method, we examine the accuracy of populating the RMS based on the IGRG−RMS source. Finally, all of the results are evaluated during 10 representative case studies for both disturbed and quiet solar activity events, as well as an additional nine coronal mass ejection (CME) storm−featured events, as proposed by Landa and Reuveni [31]. As such, we utilized the north–east–up (NEU) position errors as the evaluation metric for comparison with various available GIM products. All NEU metrics were calculated by the "gLab" software [32] augmented with one−day predicted CODE GIMs and their corresponding RMS predicted maps, as a single−frequency (L1 = 1575.42 MHz) positioning correction input.

## 2. Methodology

In order to predict the RMS TEC maps 24 h in advance, we followed [31]'s methodology, utilizing the DMD approach, which was first introduced by SCHMID [33]. The DMD is a technique rooted in fluid dynamics that harnesses data analysis to reveal hidden patterns in complex data with various dimensions [33]. It utilizes a combination of proper orthogonal decomposition (POD) and singular value decomposition (SVD) to efficiently extract the spatiotemporal structures of high−dimensional data [34,35]. The DMD method

approximates the best linear transformation, which transforms a system state at time $t$ to a system state at a time $t + 1$. More formally, the transformation $A$ can be approximated by minimizing the flowing expression:

$$A = \arg\min_{A} ||X' - AX||_F = X'X^{\dagger} \tag{1}$$

Here, $X'$ and $X$ represent matrices of system states stacked together as column vectors of snapshots corresponding to the times $t \in \{2, 3, 4, ..., n + 1\}$ and $t \in \{1, 2, 3, ..., n\}$, respectively. In our case, the matrices $X'$ and $X$ contain the TEC RMS maps reshaped into column vectors and stacked together, forming the matrices $X'$ and $X$. The $^{\dagger}$ denotes the pseudo−inverse, and $|| \cdot ||_F$ represents the Frobenius norm. One can utilize matrix $A$ to predict one time step of the system state in advance as follows:

$$\tilde{x}_{t+1} = \tilde{A}\tilde{x} \tag{2}$$

Based on the origin time step, repeating such predictions, results in a target timespan forecast period. For a more detailed description of the methodology, see [31].

## 3. Data and Forecast

In order to validate the suggested methodology, we selected multiple available ionospheric correction models, namely, the IGS final products (IGSG), Klobuchar model [8], Wuhan University (WHU) rapid GIM products, JPL rapid GIM products [36], and ESA rapid GIM products, as comparison models. Similar to previous studies that refer to the IGSG as the reference model [20,31,37], we also consider the IGSG final product as the baseline model for both the comparison and RMS data. In addition, we utilize the IGS rapid product (IGRG) RMS data to examine the methodology from a more practical perspective. Therefore, we select the IGSG and IGRG products with a 2 h candidate GIM source of RMS data for constructing a suitable data set for the DMD. In the same manner, we select the publicly available 24 h predicted GIMs with 2 h candidate data, provided by the CODE analysis center, as the target source. Based on our previous work [31], the selected products in this study were chosen to cover three main solar periods: solar cycle 24 disturbance activities (during 2014), solar cycle 24 quiet activities (during 2014), and additional CME events that occurred between 2015 and 2019 (Table 1).

**Table 1.** Table of case studies presenting selected solar activity dates and events, including CME, quiet, and disturbance periods with the corresponding Kp indices and X-ray magnitudes.

| Evaluated Case Study | | | | | |
|---|---|---|---|---|---|
| **Quiet Events** | **X-ray Class** | **Disturbance Events** | **X-ray Class** | **CME Record** | **Max Kp Index** |
| 23 August 2014 | C6.0 | 25 February 2014 | X4.9 | 17 March 2015 | 8− |
| 25 January 2014 | C1.0 | 24 October 2014 | X3.1 | 22 June 2015 | 8+ |
| 15 February 2014 | C2.3 | 10 June 2014 | X2.2 | 23 June 2015 | 8− |
| 26 March 2014 | C1.5 | 27 October 2014 | X2.0 | 8 September 2017 | 8+ |
| 9 April 2014 | C2.1 | 26 October 2014 | X2.0 | 28 September 2017 | 7− |
| 29 May 2014 | C1.4 | 20 December 2014 | X1.8 | 25 August 2018 | 4+ |
| 23 June 2014 | C1.0 | 7 November 2014 | X1.6 | 26 August 2018 | 7+ |
| 22 July 2014 | B2.8 | 22 October 2014 | X1.6 | 20 August 2018 | 6 |
| 20 September 2014 | C1.1 | 10 September 2014 | X1.6 | 1 September 2019 | 5+ |
| 11 October 2014 | C1.8 | 25 April 2014 | X1.3 | −/−/− | − |

The GIM products used in this study were sourced from ftp://gdc.cddis.eosdis.nasa.gov (accessed on 10 June 2023) FTP server, until 2020, and from https://cddis.nasa.gov/ (accessed on 10 June 2023) with HTTPS/FTP−SSL access. The available daily IONEX files consist of 13 or 25 GIMs at a 2 h or 1 h timespan resolution, respectively. It should be noted that the last GIM of each day (the thirteen/twenty−fifth GIM) overlaps with the first GIM

of the next day. The matrices' GIM representations have $71 \times 73$ dimensions and contain scaled vTEC values, along with their associated RMS values, which correspond to $(2.5°, 5°)$ latitude and longitude resolutions, respectively [29,37]. An example of a typical IGS vTEC RMS map is shown in Figure 1 (Up−A), while Figure 1 (Bottom−B) depicts an IGR vTEC RMS map example provided by the IGS data center.

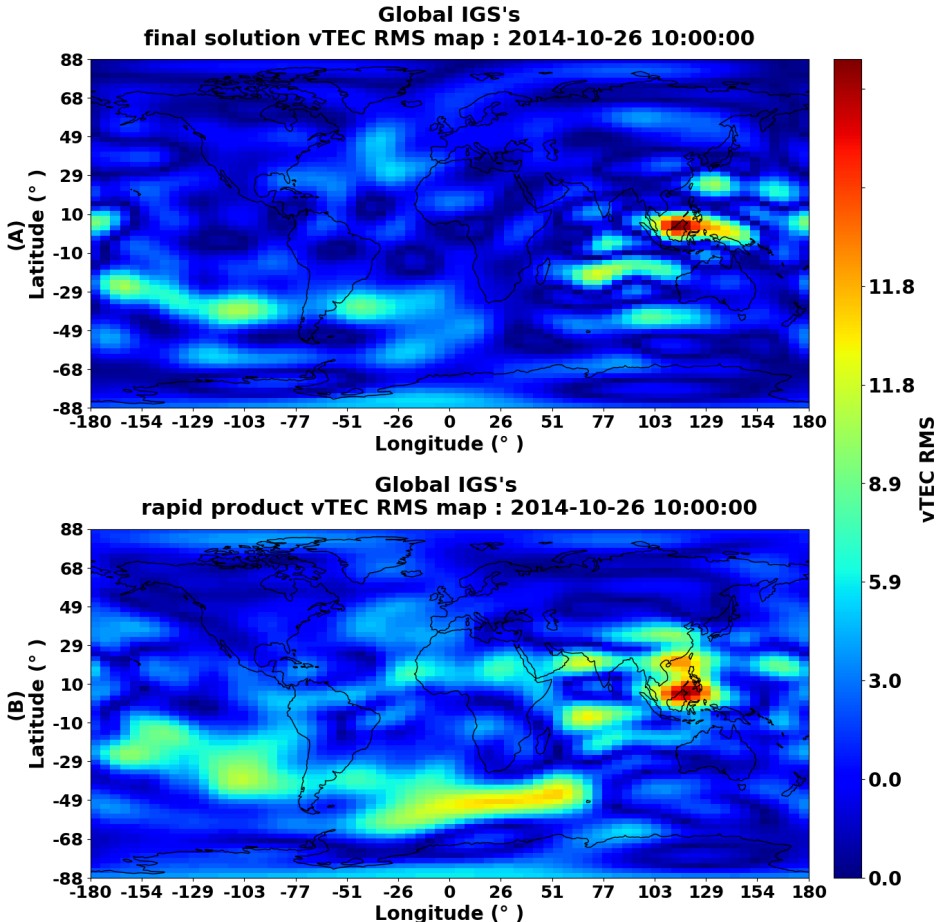

**Figure 1.** vTEC RMS GIMs: (**A**) Final solution RMS (IGSG−RMS) product dated 26 October 2014 at 10:00 and sourced from IGS; (**B**) rapid product RMS (IGRG−RMS) dated 26 October 2014 at 10:00 and sourced from IGS.

### 3.1. Preprocessing

The use of DMD as a vTEC RMS GIMs predictor requires constructing a suitable and well-organized data set. Specifically, it requires a data set that is structured as a matrix and consists of system state snapshots evolving over time, stacked as column vectors. For this purpose, we refer to each vTEC RMS GIM with a 2 h cadence system snapshot.

As a prerequisite, we select daily IONEX files, sourced from IGS, encompassing a complete year of snapshots for each of the case studies listed in Table 1. This results in 365 IONEX files for each year between 2013 and 2019, except for 2016. For each IONEX file, we select only the RMS GIM part, resulting in a set of 13 RMS maps with 2 h intervals. If the IONEX file contains 25 RMS GIMs, we select only 13 RMS GIMs, such that the timespan interval between them is equal to 2 h. Then, we exclude the last (thirteenth) RMS GIM, which coincides with the first one in the subsequent day's file. The remaining 12 RMS maps constitute the system snapshots, where each snapshot corresponds to an RMS GIM acquired every two hours. We reshape each RMS GIM snapshot into a column vector form and merge them together, forming a daily system matrix with 12 columns of snapshots. Specifically, each original $71 \times 73$ RMS map is transformed into a column vector of the dimensions $5183 \times 1$ and joined to produce a $5183 \times 12$ daily 2 h snapshot matrix.

Finally, all daily snapshot matrices is stacked into a single data set matrix that represents a dynamical system of the snapshots acquired at this 2 h cadence. The general data matrix is created entirely from the selected data range and spans a total of 2190 (365 × 6) days, resulting in a matrix with the dimensions 5183× 26,280 (26,280 = 2190 × 12). Figure 2 illustrates that process for the years 2013 and 2014 range.

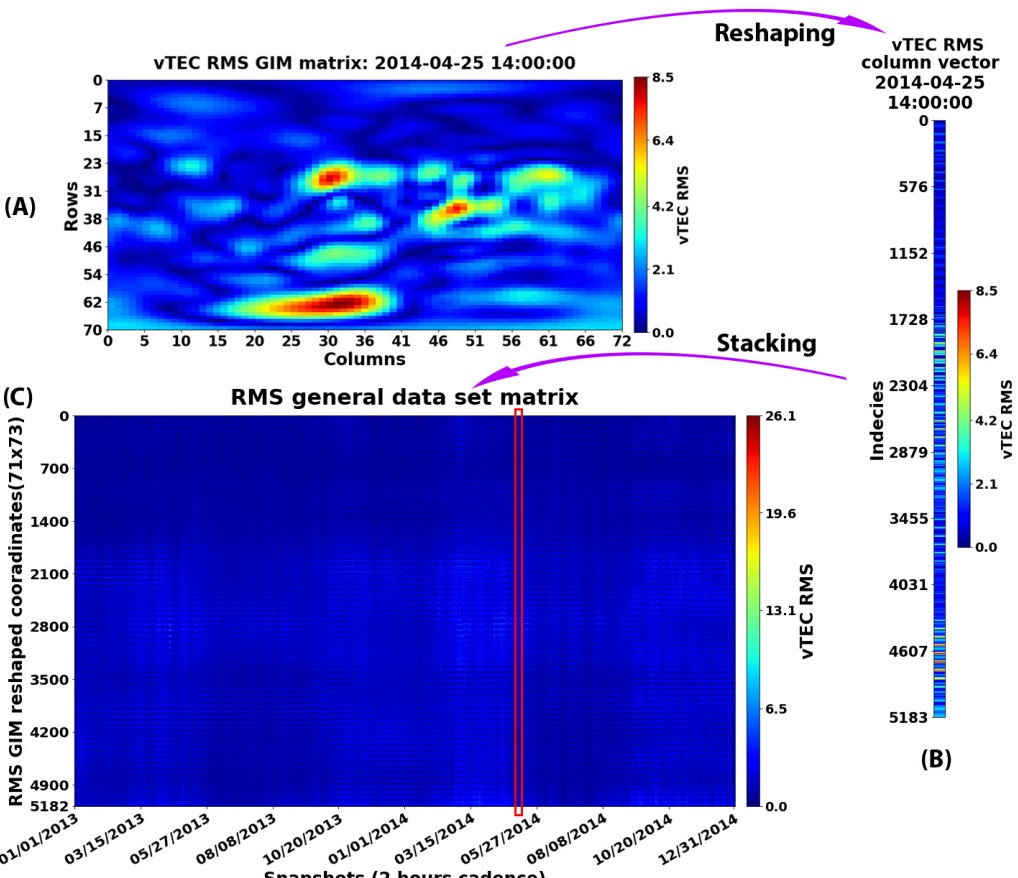

**Figure 2.** Illustration of preprocessing and constructing the DMD data set matrix: (**A**) 71 × 73 (dim) vTEC RMS GIM sourced from IGS and dated 25 April 2014 at 14:00; (**B**) RMS map reshaped into a column vector; (**C**) final daily stacked column vectors data set matrix, covering system snapshots of years 2013 and 2014.

*3.2. RMS GIM Predictions and Assimilation*

To generate daily predictions of RMS GIMs using the DMD algorithm, we first select a sub-matrix out of the general matrix to include a history of *h* snapshots prior to the target prediction RMS GIM. As described in previous work, we select *h* to be 1440, which corresponds to approximately 4 months of historical data. Then, we apply the DMD methodology to estimate the transition matrix *A* using the selected sub-matrix. In turn, the approximated transformation *A*, along with the last RMS map snapshot in the sub-matrix is used to predict the following 24 RMS maps in advance, as described in Equation (2). For each case study in Table 1, we apply the DMD algorithm using the corresponding data set matrix and utilize the yielded transformation *A* to predict the first 24 h with a 2 h time step.

Once we obtain the predicted 24 h of RMS GIMs, we assimilate them within the corresponding one-day prediction CODE IONEX file, which is selected based on the different case studies in Table 1. The assimilation process populates the RMS values according to the IONEX file format [29].

## 4. Results

Using the methodology described above, we conducted a comprehensive assessment of the 24 h periods with 2 h candidate predicted RMS GIMs assimilated within CODE-predicted IONEX files in various scenarios. Specifically, we examined 10 case studies characterized as disturbed solar weather (resulting from high solar flare activity, emitting X-ray radiation and extreme UV, which strongly affected the F2 Ionospheric layer), 10 case studies characterized as quiet solar weather, and 9 case studies during CME events (as shown in Table 1). First, we applied the DMD with a relevant RMS data set matrix, derived from IGSG GIMs, to predict 24 h with 2 h candidates of RMS maps for each case study. Next, we assimilated the predicted RMS GIMs with their corresponding CODE IONEX (IGSG-RMS-CODE) files and evaluated the new model with an NEU position error metric, based on the acquired L1 single-frequency position solutions, utilizing the open-source GNSS-Lab Tool software (gLAB v5.5.1) [32]. The gLAB software was fed with relevant raw receiver-independent exchange format RINEX files and the IGSG final, Klobuchar, WHU, JPL, and ESA GIM rapid products' ionospheric correction models. The RINEXes were recorded by a single GNSS station receiver, located at 32.77899 latitude and 35.02298 longitude with a 225.1 m elevation. Figures 3–5 show the average NEU errors over 24 h and the NEU error statistics for different models compared to our IGSG-RMS-CODE assimilated model, throughout the selected case studies. In addition, Table 2 summarizes the results statistics of the absolute NEU errors.

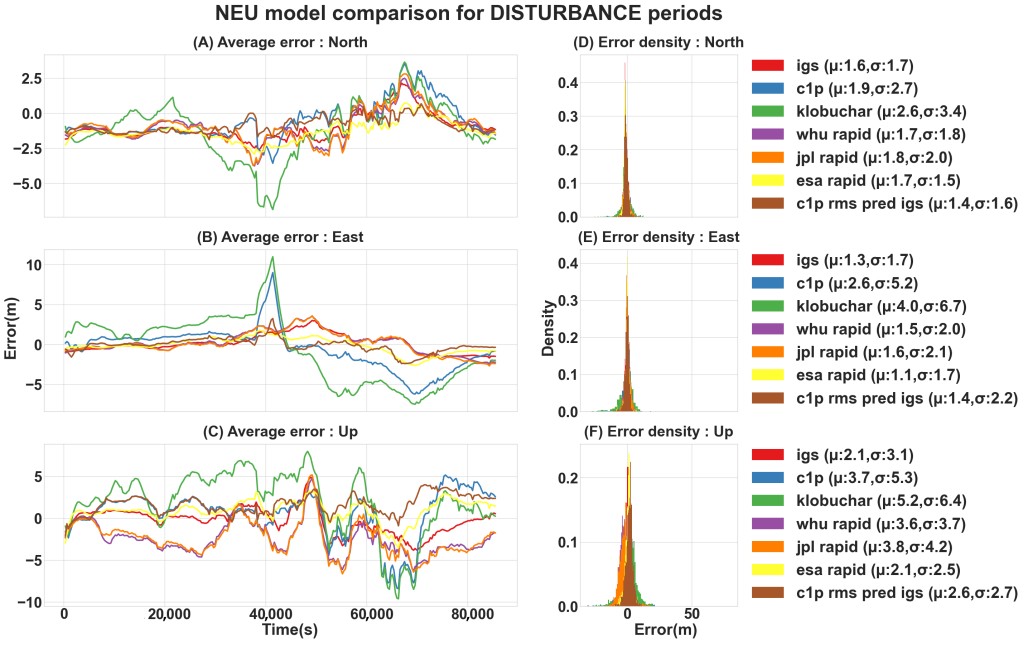

**Figure 3.** NEU error [m] comparison of disturbance case studies: (**A–C**) show the average north, east, and up errors over 24 h, respectively; (**D–F**) show the total north, east, and up error distributions, respectively. (The statistics are presented for NEU absolute values).

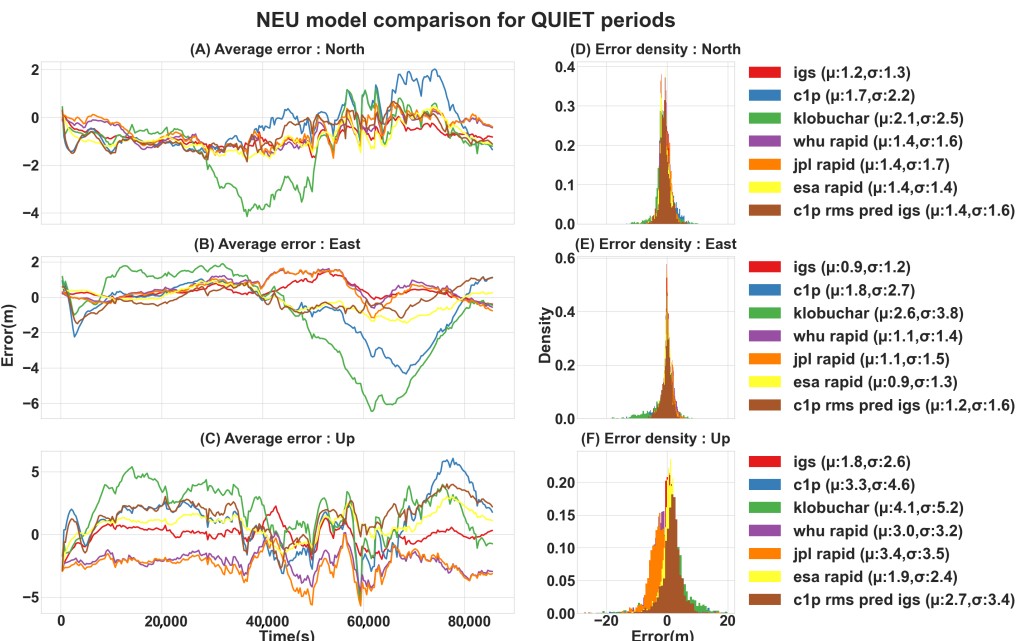

**Figure 4.** NEU error [m] comparison of quiet case studies: (**A**–**C**) show the average north, east, and up errors over 24 h, respectively; (**D**–**F**) show the total north, east, and up error distributions, respectively. (The statistics are presented for NEU absolute values).

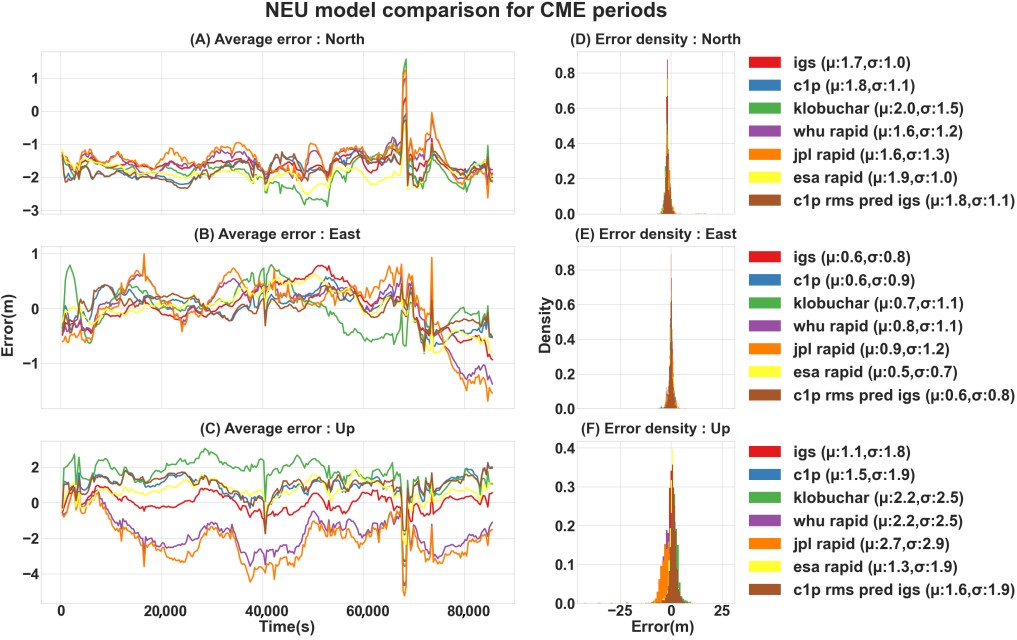

**Figure 5.** NEU error [m] comparison of CME case studies. (**A**–**C**) show the average north, east, and up errors over 24 h, respectively; (**D**–**F**) show the total north, east, and up error distributions, respectively. (The statistics are presented for NEU absolute values).

**Table 2.** Selected case studies' absolute NEU error [m] statistics, describing different solar weather periods and error types.

| | | | | | | | | | **DMD (IGSG-RMS-CODE)** |
|---|---|---|---|---|---|---|---|---|---|
| **Period** | **Stat** | **Error** | **IGS** | **C1P** | **Klobuchar** | **WHU** | **JPL** | **ESA** | |
| Disturbed | AVG | North | 1.6 | 1.9 | 2.6 | 1.7 | 1.8 | 1.7 | 1.4 |
| | | East | 1.3 | 2.6 | 4.0 | 1.5 | 1.6 | 1.1 | 1.4 |
| | | Up | 2.1 | 3.7 | 5.2 | 3.6 | 3.8 | 2.1 | 2.6 |
| | STD | North | 1.7 | 2.7 | 3.4 | 1.8 | 2.0 | 1.5 | 1.6 |
| | | East | 1.7 | 5.2 | 6.7 | 2.0 | 2.1 | 1.7 | 2.2 |
| | | Up | 3.1 | 5.3 | 6.4 | 3.7 | 4.2 | 2.5 | 2.7 |
| Quiet | AVG | North | 1.2 | 1.7 | 2.1 | 1.4 | 1.4 | 1.4 | 1.4 |
| | | East | 0.9 | 1.8 | 2.6 | 1.1 | 1.1 | 0.9 | 1.2 |
| | | Up | 1.8 | 3.3 | 4.1 | 3.0 | 3.4 | 1.9 | 2.7 |
| | STD | North | 1.3 | 2.2 | 2.5 | 1.6 | 1.7 | 1.4 | 1.6 |
| | | East | 1.2 | 2.7 | 3.8 | 1.4 | 1.5 | 1.3 | 1.6 |
| | | Up | 2.6 | 4.6 | 5.2 | 3.2 | 3.5 | 2.4 | 3.4 |
| CME | AVG | North | 1.7 | 1.8 | 2.0 | 1.6 | 1.6 | 1.9 | 1.8 |
| | | East | 0.6 | 0.6 | 0.7 | 0.8 | 0.9 | 0.5 | 0.6 |
| | | Up | 1.1 | 1.5 | 2.2 | 2.2 | 2.7 | 1.3 | 1.6 |
| | STD | North | 1.0 | 1.1 | 1.5 | 1.2 | 1.3 | 1.0 | 1.1 |
| | | East | 0.8 | 0.9 | 1.1 | 1.1 | 1.2 | 0.7 | 0.8 |
| | | Up | 1.8 | 1.9 | 2.5 | 2.5 | 2.9 | 1.9 | 1.9 |

The spanning header of the table reads: **Cases Studies Statistics Analysis of Absolute NEU Errors [Meter]**

Furthermore, we assessed the assimilated model from a more practical perspective. Specifically, we utilized the IGRG−RMS maps with the DMD framework in order to generate a 24 h prediction of RMS with two−hour intervals based on the IGSG rapid product. The use of IGSG final products as an RMS training source is substantially impractical, due to the latency (from 13 to 20 days) in the availability of the products. In contrast, the IGSG rapid product (IGRG) constitutes a more suitable source for RMS predictions. Its data are available every day, with a delay of approximately 17 h from the end of the previous observation day. This means that 24 h of RMS predictions construct RMS assimilation for the last five hours of the one−day predicted CODE GIMs (C1P). Therefore, we applied the aforementioned DMD forecasting methodology with the IGRG−RMS maps and sequentially assimilated the resulting RMS maps with the corresponding one−day predictions CODE products (IGRG−RMS−CODE). Finally, we compared the IGRG−RMS−CODE model with the IGS final solution and the IGSG−RMS−CODE model regarding NEU position errors. Figures 6–8 show the average NEU errors over 24 h and the NEU error statistics of the IGS and IGSG−RMS−CODE assimilated models compared to the IGRG−RMS−CODE assimilated model throughout various case studies. In addition, Table 3 summarizes the results statistics of the absolute NEU errors.

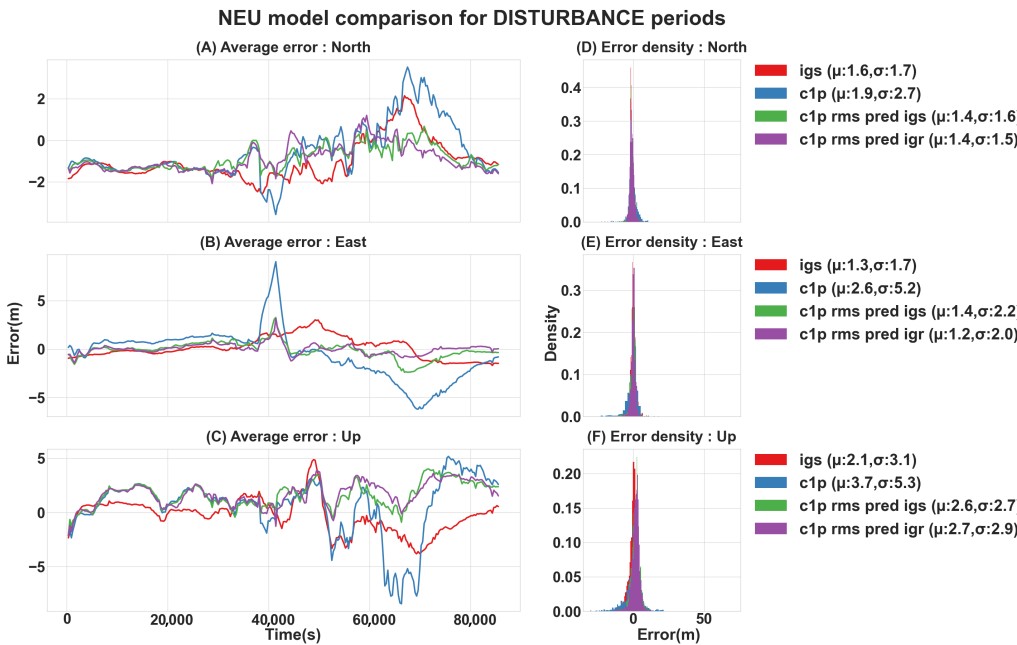

**Figure 6.** IGRG−RMS−CODE NEU error [m] comparison of disturbance case studies: (**A**–**C**) show the average north, east, and up errors over 24 h, respectively; (**D**–**F**) show the total north, east, and up error distributions, respectively. (The statistics are presented for NEU absolute values).

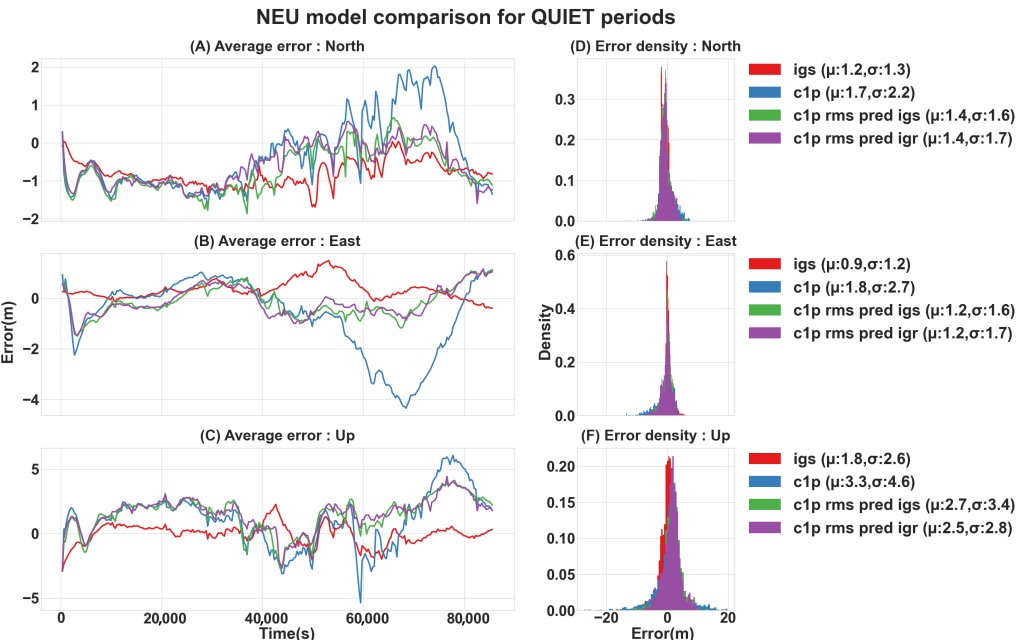

**Figure 7.** IGRG−RMS−CODE NEU error [m] comparison of quiet case studies: (**A**–**C**) show the average north, east, and up errors over 24 h, respectively; (**D**–**F**) show the total north, east, and up error distribution, respectively. (The statistics are presented for NEU absolute values).

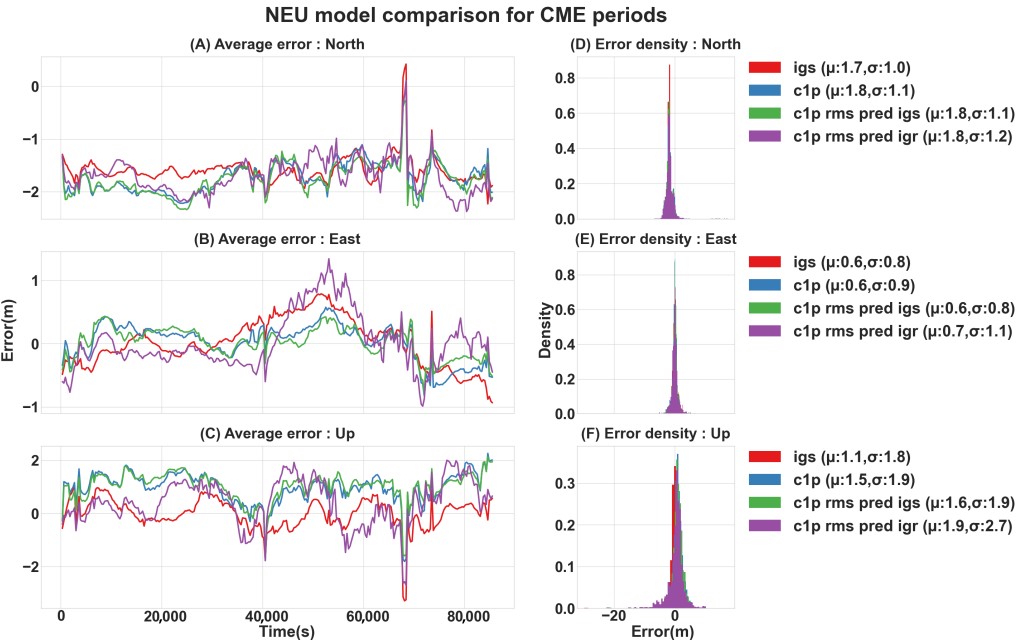

**Figure 8.** IGRG−RMS−CODE NEU error [m] comparison in CME case studies: (**A**–**C**) show the average north, east, and up errors over 24 h, respectively; (**D**–**F**) show the total north, east, and up error distribution, respectively. (The statistics are presented for NEU absolute values).

**Table 3.** Absolute IGRG−RMS−CODE model NEU error [m] statistics of the case studies, sorted by different solar activity periods and error types.

| Case Studies Absolute IGRG−RMS−CODE Model NEU Errors [Meter] Statistics | | | | | | |
|---|---|---|---|---|---|---|
| **Period** | **Stat** | **Error** | **IGS** | **C1P** | **DMD (IGSG−RMS−CODE)** | **DMD (IGRG−RMS−CODE)** |
| Disturbed | AVG | North | 1.6 | 1.9 | 1.4 | 1.4 |
| | | East | 1.3 | 2.6 | 1.4 | 1.2 |
| | | Up | 2.1 | 3.7 | 2.6 | 2.7 |
| | STD | North | 1.7 | 2.7 | 1.6 | 1.5 |
| | | East | 1.7 | 5.2 | 2.2 | 2.0 |
| | | Up | 3.1 | 5.3 | 2.7 | 2.9 |
| Quiet | AVG | North | 1.2 | 1.7 | 1.4 | 1.4 |
| | | East | 0.9 | 1.8 | 1.2 | 1.2 |
| | | Up | 1.8 | 3.3 | 2.7 | 2.5 |
| | STD | North | 1.3 | 2.2 | 1.6 | 1.7 |
| | | East | 1.2 | 2.7 | 1.6 | 1.7 |
| | | Up | 2.6 | 4.6 | 3.4 | 2.8 |
| CME | AVG | North | 1.7 | 1.8 | 1.8 | 1.8 |
| | | East | 0.6 | 0.6 | 0.6 | 0.7 |
| | | Up | 1.1 | 1.5 | 1.6 | 1.9 |
| | STD | North | 1.0 | 1.1 | 1.1 | 1.2 |
| | | East | 0.8 | 0.9 | 0.8 | 1.1 |
| | | Up | 1.8 | 1.9 | 1.9 | 2.7 |

## 5. Discussion

The NEU assessments of the predicted RMS maps demonstrated that the DMD algorithm can sufficiently forecast accurate global ionospheric vTEC RMS maps, 24 h ahead with a 2 h time interval. Furthermore, the evaluations suggested that the predicted RMS maps are appropriate for integration with other ionospheric products, such as C1P (CODE) maps.

In addition, the evaluation of NEU error distributions for all solar activity case studies (quiet, disturbed, and CMEs), presented in Table 3, yielded two key findings. The first one is utilizing the IGSG product's RMS maps as the source of RMS prediction with the DMD framework, and its assimilation with C1P ionospheric forecast maps projects NEU error distribution improvement across almost all solar activity case studies and other models as well. To start, the IGSG−RMS−CODE model shows higher statistical results than its base model (without RMS contribution), the C1P (CODE), in the disturbed and quiet case studies, and it remains the same in the CMEs case studies. Moreover, the presented model showed frequent improvement over the JPL rapid model, but still, these were comparably close. In addition, it achieved similar statistical scores to other rapid models, with alternating improvements and degradation. For example, in the east error during quiet periods, the WHU model presented a mean value of 1.1 m and a standard deviation of 1.4 m, whereas the IGSG−RMS−CODE presented a higher mean error value of 1.2 m and a higher standard deviation error of 1.6 m. In contrast, the up error during the CME period of the WHU model showed a mean value of 2.2 m and a standard deviation of 2.5 m, while the IGSG−RMS−CODE showed a lower mean error value of 1.6 m and a lower standard deviation error of 1.9 m. Note that the ESA rapid model has higher mean and standard deviation error statistics than the IGSG−RMS−CODE model in almost all case studies (see Table 2). Secondly, despite the inherent inaccuracy in the IGRG product's RMS maps, due to their rapid availability, the IGRG−RMS−CODE model achieved a very similar statistical score to the ones based on the IGSG RMS maps. In some cases, the achieved scores were better (i.e., lower mean and standard deviation errors) than the IGSG−RMS−CODE model. For example, in the up error of the quiet periods, the IGSG−RMS−CODE resulted in an average error value of 2.7 m and a standard deviation of 3.4 m, whereas the IGRG−RMS−CODE model resulted in an average error value of 2.5 TECu and a standard deviation of 2.8 TECu. However, overall, both of the presented models have almost identical scores (see Table 3).

The results shown here with the IGRG−RMS−CODE model enhance the ability of future planning for applications that rely on L1 single−frequency positioning solutions with ionospheric correction models. Specifically, the availability of IGSG rapid products is around ∼17 h, consequently, the IGRG−RMS−CODE ionospheric correction model can be used to examine the positioning accuracy for predicted ionospheric maps up to 5 h ahead.

## 6. Conclusions

In this study, we focused on generating 24 h of vTEC RMS GIM prediction, utilizing the previously suggested DMD methodology. To support this effort, we utilized two available vTEC GIM RMS sources, namely IGSG and IGRG products. The IGSG RMS data were used as a reference frame to establish whether the proposed methodology is suitable for RMS map predictions and assimilation as additional information for other available models, specifically, the C1P (CODE) product. In turn, we examine the presented approach from a more practical perspective, utilizing the IGRG product. Due to its rapid availability, its RMS maps constitute a suitable source for future assimilation with a C1P predicted product, thus forming the C1P−RMS model for advanced time periods. We performed a comprehensive evaluation with 29 different case studies, such that 10 cases were chosen during quiet solar activity periods, 10 cases were chosen during disturbed solar activity periods, and an additional 9 cases were chosen during CME storm events. All of the evaluations were made using the NEU metric and compared to the IGS, C1P, Klobuhcar, WHU, JPL, and ESA rapid models. We successfully applied the DMD model with an ionospheric vTEC GIM RMS maps data set and constructed a 24 h global ionospheric vTEC RMS map forecast with a 2 h candidate. The IGSG RMS prediction−based model (IGSG−RMS−CODE) indicated noticeable improvement over the base assimilation model and set comparable results to other models. Moreover, the practical attempt showed nearly identical statistical improvement to the IGSG−RMS−CODE model and made it possible to evaluate the positioning accuracy up to 5 h ahead. These findings demonstrate the impact of RMS

information for L1 single−frequency positioning solutions using ionospheric correction models and highlight that the ability to predict RMS maps as complementary information for ionospheric vTEC products can set a path for real−time PPP applications and even enable the utilization of predicted ionospheric disturbance for planning future missions that rely on PPP measurements.

**Author Contributions:** All authors have made significant contributions to the manuscript. V.L. processed the GPS−TEC and EUV data, designed and implemented the DMD algorithm development, wrote the main manuscript, and prepared the figures and tables; Y.R. conceived and designed part of the algorithm, analyzed the data and results, and is the main author who developed and revised the manuscript. All authors have read and agreed to the published version of the manuscript.

**Funding:** This research was funded in part by the Israel Ministry of Defense (MAFAT), grant number: 4441263780, and in part by the Ariel University Data Science and Artificial Intelligence Center, grant number: RA22−235.

**Data Availability Statement:** The data presented in this study are contained within the article, in Section 3. The script files and codes are available at https://github.com/vladlanda/Toward-Real-Time-Single-Frequency-Precise-Point-Positioning (accessed on 25 June 2023).

**Conflicts of Interest:** The authors declare no conflict of interest. The funders had no role in the design of the study; in the collection, analyses, or interpretation of data; in the writing of the manuscript; or in the decision to publish the results.

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
