# Peer review of "Toward Real-Time GNSS Single-Frequency Precise Point Positioning Using Ionospheric Corrections"

_remotesensing, doi:10.3390/rs15133333_

Round 1

Reviewer 1 Report

I suggest that the current form  of this paper is suitable for publication. 

Author Response

File attached

Reviewer 2 Report

In this contribution, the authors proposed global ionospheric vTEC RMS map forecasts to improve the single-frequency precise point positioning. However, major revisions may be necessary for the possible publication.

Major concerns:

[1]. Missing detailed descriptions on single-frequency precise point positioning (SF-PPP) processing strategies, including the observations utilized, parameter estimation strategy, and the error correction methods. Listing a table might be clear for readers.

[2]. As is widely known, in SF-PPP, ionospheric delays embedded in the code and phase observations can be corrected directly or constrained by ionospheric pseudo-observations using a priori ionospheric delays obtained from GIM or other data sources. See the following DOI: 10.1007/s00190-019-01311-4

So, which model is utilized in this research? And the most important issue is, how RMS maps improve the positioning accuracy of real-time SF-PPP? Please specify.

[3] As is widely known, real-time TEC models are broadcast by some analysis centers (such as CNES and CAS) under the IGS Real-Time Service (RTS) framework.

See:

https://doi.org/10.5194/essd-13-4567-2021

https://doi.org/10.1186/s43020-021-00050-2

A predicting-plus-modeling approach is used by CAS for the computation of RT-GIM. CAS RT-GIM is generated with multi-GNSS real-time data streams. The broadcasted CAS RT-GIM is computed by the weighted combination of real-time VTEC spherical harmonic coefficients and predicted ionospheric information. CNES also uses a spherical harmonic model for global VTEC representation. Spherical harmonic coefficients are computed by means of a Kalman filter and simultaneous STEC from 100 stations of the real-time IGS network. Some other analysis centers are also broadcasting real-time VTEC models. Studies have shown that the performance of SFPPP can be significantly improved by these models, see:

10.1007/s10291-018-0802-2

10.1088/1361-6501/ac0a0e

So, the authors should compare their SF-PPP results with solutions using these real-time VTEC models.

Moderate editing of English language required

Author Response

File attached

Reviewer 3 Report

The authors propose a model for ionospheric TEC prediction. Ionospheric TEC prediction is an interesting topic for the scientific community as shown in a recent review paper.

1) A. Siemuri, K. Selvan, H. Kuusniemi, P. Valisuo and M. S. Elmusrati, "A Systematic Review of Machine Learning Techniques for GNSS Use Cases," in IEEE Transactions on Aerospace and Electronic Systems, vol. 58, no. 6, pp. 5043-5077, Dec. 2022, doi: 10.1109/TAES.2022.3219366.

The introduction section needs editing. Given that TEC forecasting and similar case studies is an interesting topic, recently there have been emerged many research works such as:

1) Kaselimi, M., Voulodimos, A., Doulamis, N., Doulamis, A. and Delikaraoglou, D., 2021. Deep recurrent neural networks for ionospheric variations estimation using gnss measurements. IEEE Transactions on Geoscience and Remote Sensing60, pp.1-15.

2) Natras, R., Soja, B. and Schmidt, M., 2022. Ensemble Machine Learning of Random Forest, AdaBoost and XGBoost for Vertical Total Electron Content Forecasting. Remote Sensing14(15), p.3547. 

 3) Gomez, A.R. and Pi, X., 2021, September. Applying Machine Learning to Predict Alaskan Ionospheric Irregularities. In Proceedings of the 34th International Technical Meeting of the Satellite Division of The Institute of Navigation (ION GNSS+ 2021) (pp. 3848-3858). 

In the last paragraph of the introduction there is a discussion on how the authors implement their idea (what tools they used, how they validate their results). The authors should consider to add a figure in the methodology section to describe the steps they follow.

In Section 2 Methodology the authors should refer what X_(t+1) and X values include (e.g., TEC RMS values, etc.)

In figures 5-8 the authors should also provide the error for the 95% confidence interval, since now the figures are small and the readers cannot see the exact values. Probably and additional table can be helpful.  

There are some errors that can be improved.

Author Response

File attached

Round 2

Reviewer 2 Report

No further comments.

Reviewer 3 Report

My comments have been addressed I recommend the paper for publication.